



# Early instrumental meteorological observations in Switzerland: 1708–1873

Yuri Brugnara[1,2], Lucas Pfister[1,2], Leonie Villiger[1,2,3], Christian Rohr[1,4], Francesco Alessandro Isotta[5], and Stefan Brönnimann[1,2]

[1]Oeschger Centre for Climate Change Research, University of Bern, Switzerland
[2]Institute of Geography, University of Bern, Switzerland
[3]Institute for Atmospheric and Climate Science, ETH Zurich, Zurich, Switzerland
[4]Institute of History, University of Bern, Switzerland
[5]Federal Office of Meteorology and Climatology MeteoSwiss, Zurich, Switzerland

**Correspondence:** Yuri Brugnara (yuri.brugnara@giub.unibe.ch)

**Abstract.** We describe a dataset of recently digitised meteorological observations from 40 locations in today's Switzerland, covering the 18th and 19th century. Three fundamental variables — temperature, pressure, and precipitation — are provided in a standard format, after they have been converted into modern units and quality controlled. The raw data produced by the digitisation, often including additional variables and annotations, are also provided. Digitisation was performed by manually typing the data from photographs of the original sources, which were in most cases handwritten weather diaries. These observations will be important for studying past climate variability in Central Europe and in the Alps, although the general scarcity of metadata (e.g., detailed information on the instruments and their exposure) implies that some caution is required when using them. The data described in this paper can be found at https://doi.pangaea.de/10.1594/PANGAEA.909141 (Brugnara, 2019).

## 1 Introduction

Past meteorological observations are fundamental to understand climate variability. While the recent variability dominated by the anthropogenic warming has been extensively studied (Stocker et al., 2014), less is known about the previous centuries, a period characterised by large regional climate oscillations related to natural forcings (Brönnimann et al., 2019b; Neukom et al., 2019).

Global climate datasets based on instrumental observations typically begin in the late 19th century, when centralised meteorological networks were well established in most developed countries. The measurement of meteorological parameters in a scientific fashion, however, has a much longer history that goes back to the 17th century (Camuffo and Bertolin, 2012).

There are several reasons why early observations have been scarcely employed in modern climate research. One of the most important concerns data quality, particularly for temperature. The lack of official standards before the creation of national weather services (NWSs) makes observations difficult to compare with modern records, especially if stations are few and far apart. Moreover, until at least the 1770s many different temperature scales, often ambiguously defined, were used and spirit of wine was commonly preferred to mercury as thermometric liquid. Another difficulty is that most of the early observations have

never been published in extenso and are only available as manuscripts scattered over thousands of archives; a recent global compilation of early instrumental records (Brönnimann et al., 2019a) found that half of the known records have not yet been digitised even as monthly means. Finally, the usability of some variables, such as pressure, has changed radically with the

introduction of new assimilation techniques in reanalysis (Compo et al., 2006; Slivinski et al., 2019), so that their importance has been recognised only recently.

In the 18th and 19th century several influential scientists — such as Jacques-Barthélemy Micheli du Crest, Jean-André Deluc, Johann Heinrich Lambert, Marc-Auguste Pictet, and Heinrich von Wild — were active in today's Switzerland and at the forefront of research in the field of meteorological measurements, followed by a lively scientific community eager to use

new and better instruments. Learned societies were founded in the main towns, where their members could present and discuss the latest scientific discoveries. Many of these societies had their own meteorological observatory or even set up a regional network of stations (Pfister, 1975; Hupfer, 2015, 2017; Pfister et al., 2019). However, very few records from the time before the creation of a NWS in 1863 have been used in modern climate research (e.g., Auchmann et al., 2012; Brugnara et al., 2015).

The project CHIMES (Swiss Early Instrumental Measurements for Studying Decadal Climate Variability) has been funded

in 2016 by the Swiss National Science Foundation to compile pre-NWS observations in Switzerland and make them available in digital format. In a first paper (Pfister et al., 2019) we described the large amount of records the we found in archives and provided the digital images of nearly all documents. The present paper describes the data that we digitised and the necessary processing to make them usable.

## 2   Methods

### 40   2.1   Digitisation strategy

Given the large amount of meteorological records found in Swiss archives (see Pfister et al., 2019), it was not possible to digitise everything within the available budget. Therefore, we had to set priorities. We selected 70 records from 40 locations according to the following criteria:

- length of record (longer records preferred);

- period covered (older records preferred);

- potential for extending or improving a nearby record;

- difficulty of the digitisation (readability, data structure);

- at least one of the core variables (temperature, pressure, and precipitation) must be measured.

Information on variables and the period covered by each digitised record is shown in Fig. 1, while Fig. 2 illustrates the

geographical distribution of the stations. Secondary variables (see Table 1) were digitised only when they did not increase significantly the time required to type the record. Qualitative descriptions of the weather were in general not digitised, with the





exception of some printed sources. It is important to mention that digital images of most of the records are freely accessible online (Pfister, 2019).

Manual typing was preferred over semi-automatic techniques such as Optical Character Recognition, given the large preva-
lence of hand-written sources. A citizen science approach (e.g., Hawkins et al., 2019) would have also been difficult because of the high heterogeneity in the structure of the documents.

The typing work was carried out by undergraduate geography and history students of the University of Bern. Each digitisation "package" — typically corresponding to about 10 hours of work or three to five years of observations, with large variability among sources — was prepared by a trained climatologist and included a template and template-specific instructions. The
packages were assigned to the students through an internet portal, which was also used by the students to upload the completed files. Each package was assigned to only one student, although some were reassigned to a different student because of quality issues (see Sect. 2.3). In total, nearly 300 packages were assigned to about 50 students over a period of two years.

## 2.2 Conversion to modern units

### 2.2.1 Historical background

Thermometers and temperature scales reached a certain degree of standardisation only in the late 18th century, more than one century after the invention of the liquid-in-glass thermometer. Until at least the 1770s many thermometers used for meteorological observations had unique characteristics and even instruments graduated with the same nominal scale could read very different temperatures in the same conditions (Knowles Middleton, 1966; Camuffo et al., 2017). Therefore, converting the earliest temperature observations into modern units requires detailed information on the thermometer construction and
calibration.

Figure 3 gives an overview of the scales used in our dataset and the periods in which they were employed. The Réaumur scale, originally defined by René-Antoine Ferchault de Réaumur in 1730, was clearly the dominant scale in Switzerland until the mid-19th century. Today's Celsius scale came into use only in the 1830s and became the prevalent scale by the 1850s.

The history of the Réaumur scale is rather complex as the principles given by Réaumur were very difficult to apply in prac-
tice; as a consequence, early Réaumur thermometers were not consistent with each other and the construction methods changed radically over time (van Swinden, 1778; Knowles Middleton, 1966; Gauvin, 2012). Aside from Réaumur, early instrumental temperature measurements in Switzerland were greatly influenced by two Genevan scientists, Jacques-Barthélemy Micheli du Crest (1690–1766) and Jean-André Deluc (1727–1817). The former proposed in 1741 a "universal" thermometer filled with spirit of wine and graduated after two fixed points: the temperature of the cellar of the Paris observatory (which Micheli du
Crest believed to be more constant than the melting point of ice) and the boiling point of water at an atmospheric pressure of 27.75 Paris inches, corresponding to 0 and 100 degrees, respectively (Micheli du Crest, 1741). The lower fixed point was changed to the melting point of ice already in 1742 (Talas, 2002), but the zero remained at the "temperate" level of the observatory's cellar, so that the melting point of ice was defined at -10.4 degrees. During the following decades, Micheli du Crest's thermometer proved to be very successful, particularly in the German-speaking Swiss cantons (central and eastern Switzerland)





and in Bavaria, whereas most French scholars were uncomfortable with Micheli du Crest's disrespect of Réaumur's principles
(Talas, 2002). Seven of the temperature records in our dataset (four in the canton of Zurich and three in the canton of Bern)
were originally in Micheli du Crest units (Fig. 3). In Schaffhausen, the physician Johann Christoph Schalch read a similar
thermometer for over 50 years, between 1794 and 1845.

Deluc published in 1772 a very influential work on meteorological instruments (Deluc, 1772), where he advocated the
superiority of mercury as a thermometric liquid and applied Micheli du Crest's ideas to a mercury thermometer. His fixed points
were the melting point of ice (zero) and the boiling point of water at a constant pressure (80 degrees). Deluc's publication gave
a fundamental push to the reformation of the Réaumur thermometer — which used spirit of wine and whose scale was based
on one fixed point — into the much better standardised mercury thermometers employed in most of Europe between the late
18th century and the mid-19th century (improperly called "Réaumur thermometers" by contemporaries). We assumed that all
records in Réaumur units in our dataset from 1778 onwards were measured with the reformed thermometer.

An additional important source of uncertainty for temperature observations is the exposure of the thermometer. Already in
the 18th century observers were aware that the thermometer should not be exposed to direct or scattered solar radiation or to
precipitation, but proper radiation screens came into use only in the second half of the 19th century (e.g., Wild, 1860). In the
first half of the 18th century, the common practice was to measure indoor next to an open window in an unheated room (e.g.,
Lambert, 1758), until comparisons with outdoor thermometers demonstrated the inadequacy of this setup (e.g., Miles, 1747).
In later years, the most popular solution was to hang the thermometer outside a north-facing window or on a north-facing wall
(e.g., Carrard, 1763). Positioning-related biases can reach several degrees Celsius and are usually corrected using a statistical
approach (e.g., Böhm et al., 2010; Brugnara et al., 2016); we plan to address this issue for some of the records in a separate
paper.

Air pressure observations involve, to some extent, smaller difficulties than temperature: the liquid used was mercury with
very few exceptions and the problem of the scale definition is much less relevant, since pressure is linearly proportional to
the height of the mercury column. Moreover, barometers could be kept indoor and were not exposed to solar radiation or
precipitation. There are still numerous sources of uncertainty, many depending on the construction of the barometer (e.g.,
Camuffo et al., 2006; Brugnara et al., 2015; Grimmer, 2019). The easiest to address is the expansion of the mercury with
temperature, which follows a linear equation (Brugnara et al., 2015). Unfortunately, even though Micheli du Crest himself
had written about the importance of measuring the temperature of the barometer (Micheli du Crest, 1758), it is rare to find a
18th century record with this information. The first Swiss stations to provide attached temperature were those taking part in
the Palatine Meteorological Society's network (Cassidy, 1985), namely Gotthard Pass from 1781 and Geneva from 1783. In
the 19th century, barometers often came with correction tables and some observers annotated only the corrected values, while
others reported also the raw readings and the attached temperature.



### 2.2.2 Temperature

The oldest temperature observations that we digitised are those of Johann Jakob Scheuchzer in Zurich, starting in 1718. However, available information on his thermometer was insufficient to attempt a conversion to modern units (most likely he employed an air thermometer, see Lenke (1964)).

120 The next temperature record in chronological order is that of Neuchatel, started in 1753 by Frédéric Moula, a mathematician who studied under Jean Bernoulli in Basel. He employed a Fahrenheit thermometer, the only one with this scale in our dataset. Moula was a mathematics professor in Berlin and Saint Petersburg before returning to Switzerland in 1752. He probably got familiar with Fahrenheit thermometers while abroad (Fahrenheit thermometers where particularly popular among German scientists in the first half of the 18th century (van Swinden, 1778)). On the first page of the first book of observations (Fig. 4),

125 Moula writes: "The thermometer used for the following observations is constructed using the Fahrenheit method. It is with mercury. The point of freezing, or of melting snow is marked at 32 degrees. That of boiling water at 212" (translated from French by the authors). This description does not correspond to the original method proposed by Fahrenheit, which used the body temperature of a healthy man for the highest fixed point (Fahrenheit, 1724). Clearly, the Fahrenheit scale used by Moula is the reformed one still in use today, hence no special correction was necessary other than the standard conversion to degrees

130 Celsius.

 The first record measured on the Réaumur scale is the work of famous polymath Johann Heinrich Lambert. He made regular meteorological observations in Chur (south-eastern Switzerland) between 1750 and 1756, although we could only digitise the observations that he published (Lambert, 1758) for one year (August 1755 to July 1756; for the previous years Lambert published only monthly extremes). Unfortunately, he did not provide details on the thermometer in that publication. The raw

135 observations for the previous years have been located but have yet to be imaged.

 A few years later, in 1760, the Economic Society (*Ökonomische Gesellschaft*) of Bern started a network of eight meteorological stations, all provided with identical thermometers with Réaumur scale. According to Pfister (1975), these thermometers were made by the French Abbé Jean Antoine Nollet, an assistant of Réaumur and his main instrument-maker. However, this information alone is not sufficient to attempt a conversion into modern units, as Nollet himself did not strictly follow Réaumur's

140 principles (Knowles Middleton, 1966; Talas, 2002). In fact, given the general lack of detailed information on how Réaumur thermometers were calibrated, we did not convert temperature measured in Réaumur degrees before 1778. Luckily, the station of Bern — the only one of the Economic Society's network for which we digitised the data — switched to a Micheli du Crest thermometer in March 1762. The network did not have a long life as observations stopped in 1770.

 To convert the Micheli du Crest scale we used the tables published by Deluc (1772) and van Swinden (1778). The corrections

145 for values inbetween the points as well as outside the range given in the table by Deluc (1772) were calculated by fitting a second degree polynomial constrained to the zero (i.e., the conversion for the zero is exactly as in the table). This translates into the following equations:

$$R_t = 10.25 + 0.897 DC \tag{1a}$$



$$R_d = -0.8 + 1.0377 R_t - 0.0026 R_t^2 \tag{1b}$$

where $R_t$ is the original or "true" Réaumur scale, $R_d$ is the reformed Réaumur scale after Deluc, and $DC$ is the Micheli du Crest scale.

We assume that all thermometers with Micheli du Crest scale underwent the same construction and calibration procedure, with one exception: the previously mentioned record of Schaffhausen by Johann Christoph Schalch. Here a thermometer made by the famous Bavarian instrument-maker Georg Friedrich Brander was used. Designated as "Brander thermometer" by Schalch, this instrument resembled a Micheli du Crest thermometer but was probably filled with mercury instead of spirit, hence requiring a different conversion function.

Schalch's record was analysed before by Gisler (1983), who concluded that Schalch's thermometer was a standard Micheli du Crest spirit thermometer. Schalch carried out parallel observations between the Brander thermometer and a thermometer with Réaumur scale from 1828 to 1842; a scatter plot of these observations is given in Fig. 5a. According to Gisler (1983): "The evaluation of the difference between simultaneous readings of Réaumur and Micheli du Crest thermometers showed clearly that the two thermometers could only be a spirit thermometer, described as Brander, and a mercury thermometer after Réaumur" (p. 36–37, translated from German by the authors). This statement is not supported by any figure or table. By analysing Schalch's parallel observations, we found that the relationship between the two scales is best represented by a linear function:

$$R_d = 11.00 + 0.91 B \tag{2}$$

where $B$ is the scale of the Brander thermometer. A linear relationship suggests that both thermometers were filled with the same liquid — most likely mercury. Besides, Eq. 2 implies that the zero of the Brander scale is at 11 degrees Réaumur, 1.4 degrees higher than defined in the Micheli du Crest scale. A possible explanation for the difference is that the zero of the Brander thermometer had risen over time because of the gradual contraction of the glass, a well-known source of systematic error particularly in mercury thermometers (Knowles Middleton, 1966; Winkler, 2009).

Brander was a correspondent of Micheli du Crest and had started to make thermometers with Micheli du Crest scale as early as 1757 (Talas, 2002); it is possible that he decided at some later point to use mercury instead of spirit, expecially after Deluc's publication and Micheli du Crest's death. It is also likely that the thermometer was already rather old when Schalch started to make regular observations in 1794, given that Brander died in 1783. In fact, in 1782 a newspaper based in Augsburg (*Augsburgische Ordinari Postzeitung* n. 178, 26 July 1782) published a short piece about an earthquake in Schaffhausen, reporting a temperature measured in that city with a "Brander thermometer", perhaps the same thermometer used by Schalch 12 years later. If so, most of the rise of the zero had probably already occurred when Schalch started his observations, meaning that Eq. 2 can be used for the entire record.

If, on the other hand, we assume that the Brander thermometer was nothing else than a standard Micheli du Crest spirit thermometer, and correct for the shifted zero, we obtain too low values at high temperatures when comparing to the parallel



observations in Réaumur (Fig. 5b). Eq. 2 produces a better agreement for the whole temperature range. Given these considerations, we adopted Eq. 2 for the conversion of the Brander scale.

Schalch's parallel record actually continues until 1845, but shows a large inhomogeneity in 1842 (Fig. 5b): in that year,
between August and September, Schalch observed only one thermometer at a time, probably because he moved them apart. He read the Brander thermometer usually at noon, the other thermometer in the morning and in the evening. In October he resumed the parallel observations, but from that moment a seasonal cycle with an amplitude of about 2 °C appears in the bias, confirming that the two thermometers were no longer next to each other.

Conversion from Deluc's (or reformed) Réaumur scale is made by multiplying by a factor of 1.25. However, one must always
keep in mind that the accuracy of the calibration and its stability over time probably varied among different instrument-makers.

Table 2 gives an overview of the conversions applied to the different instruments mentioned in this section.

### 2.2.3 Pressure

Our strategy for the temperature correction of barometer readings was to prefer a modern correction (WMO, 2008) over the corrected values provided by the observers. The latter was used only if the attached temperature was not provided. The
195 converted data are accompanied by metadata that indicate, for each observation, whether the correction was calculated by us, by the observer, or whether no correction was applied. In addition, we applied a gravity correction to all pressure observations (WMO, 2008).

Before the introduction of the metric system, the standard unit for barometers in Switzerland was the Paris inch (*pouce*), corresponding to 27.07 mmHg, and the Paris line (*ligne*), corresponding to 1/12 of a Paris inch. The oldest record of Scheuchzer
was reported in British imperial inches (Fig. 3), corresponding to 25.4 mmHg, because it had been converted for publication in a British journal (Derham, 1709). Also a few years (1839–1843) of the record of Marschlins are compatible with a scale in British inches, although we did not find any metadata about the barometer. A few more records in the 1830s and 1840s (Einsiedeln, Gottstatt, and Utzenstorf, the latter two by the same observer) have unrealistically high pressure values, possibly because an unknown length unit different from the Paris inch was used.
The conversion from millimetres to hectopascals follows from the hydrostatic equation (WMO, 2008; Brugnara et al., 2015).

### 2.2.4 Precipitation

Precipitation was measured by only a small fraction of observers in the early instrumental era. Even though the oldest record of Scheuchzer dates back to 1708, there are never more than two stations measuring precipitation at the same time before the 1860s (Fig. 3). The units adopted were Paris lines and, more recently, millimetres (Scheuchzer's earliest record was published
in British inches).



### 2.2.5 Time

The time of observation usually refers to local solar time. In the common data format all times are converted to Greenwich Mean Time (GMT). Some observers, particularly in remote locations, continued to use sundials after a standard time (Bern time, GMT+00:30) was introduced in Switzerland in 1848 (Wild, 1862). Given that the difference between mean local solar time and Bern time is less than 12 minutes anywhere in Switzerland, we assumed mean local solar time for every record.

Qualitative time entries such as "afternoon" were not converted. In this case, the time is missing in the data files and the original time entry is provided in a dedicated column. Times given as "sunrise" or "sunset" were converted using the R package suncalc version 0.5.0 (Thieurmel and Elmarhraoui, 2019).

### 2.3 Data quality assessment

For the quality control of the digitised observations we used a combination of manual checks and automatic tests implemented in the R package dataresqc (Brönnimann et al., 2018; Brugnara et al., 2019). The digitised sheets were first checked visually to make sure that the instructions were followed correctly by the student, that all the data were typed and that there were no obvious systematic mistakes. Problematic packages (about 1 in 10) were reassigned to another student with improved instructions. The reassignment allowed us to estimate the error rate of the digitisation, by comparing the columns not affected by evident systematic errors. Assuming that each and every value is digitised correctly by at least one student, we obtain an average error rate of 1.5% for temperature (for a sample of 46,395 observations), 2.1% for pressure (31,399 observations), and 1.4% for precipitation (1,165 observations). Pressure is affected by more digitisation errors than other variables because pressure readings usually contain more digits. Our estimates are rather pessimistic given that the packages that were reassigned were usually difficult ones. For printed sources (26,245 observations) the average error rate was 0.6%, whereas for handwritten sources it rose to 2.3% (52,714 observations).

The software dataresqc offers simple statistical and logical tests to detect suspicious values that are probable digitisation errors (typos). It allowed us to detect and correct nearly 3,000 typos for pressure and nearly 1,000 typos for temperature, corresponding to about 1 in 5 estimated typos for pressure and 1 in 13 for temperature. The remaining suspicious values that were not digitisation errors were flagged in the final data format. The fraction of flagged data amounts to 0.2%, corresponding to 3,832 values.

### 3 Data format

The dataset (Brugnara, 2019) is provided in the Station Exchange Format (SEF), a standard format for the exchange of rescued climate data recently defined by the Copernicus Data Rescue Service (Brönnimann et al., 2018) with the aim of facilitating ingestion into global repositories. Each file contains data and metadata for one variable and one observer in a tab-separated value (tsv) structure. Standard (i.e., available in all files) metadata include geographical coordinates, the name of the observer,



and the type of corrections for pressure. To increase data traceability, a column in the SEF files contains the values in the original units and the original time.

A total of 187 SEF files are available: 101 for temperature, 72 for pressure, and 14 for precipitation. Figure 6 shows an example of a SEF file. The filenames follow a standard structure composed by the following fields: project name, station code
(as in Fig. 2), station name, observer, starting and ending date, variable code. In some cases, multiple thermometers and/or barometers were read by the same observer: here the suffix "bis" or "ter" is added to the filename to distinguish between the different instruments. Redundant daily averages (i.e., averages of digitised sub-daily observations) are not provided in SEF.

We also provide a R data file containing a data frame with all digitised material, including additional variables that have not been converted into SEF, as well as comments from the student who typed the data (e.g., notes on readability). The structure
of the data frame is similar to the one of the SEF, with one observation time for each line, and it is explained in detail in an accompanying documentation file. Note that the data for the additional variables have not been quality-controlled.

## 4 Conclusions

A large amount of early instrumental observations from Switzerland were digitised and converted to modern units. A quality control procedure was applied mainly to detect and correct digitisation errors.
The records have various lengths, some spanning several decades, and can potentially be merged in the future with modern records to build 200 or even 250 years long daily temperature and pressure series for many Swiss cities (e.g., Aarau, Bern, Schaffhausen, St Gall, Zurich) with few gaps. The high station density should facilitate the statistical correction of biases and inhomogeneities, which can be very large due to the lack of standards and for which metadata are usually insufficient.

The dataset is also important for the study of extreme events in the pre-industrial era. The most important variables have been
converted to a standard ASCII format that is intended to facilitate data ingestion into global public repositories and consequent use by the scientific community.

## 5 Data availability

The data described in this paper are available on PANGAEA: https://doi.pangaea.de/10.1594/PANGAEA.909141 (Brugnara, 2019). The data are also incorporated into the Euro-Climhist database (Pfister et al., 2017).

*Author contributions.* YB wrote the manuscript, organised the digitisation, converted the data and performed the quality control. LP and LV carried out the archive work, collected metadata, and organised the digitisation. SB, CP, and FI led the project and contributed to the writing of the manuscript.

*Competing interests.* The authors declare that they have no competing interests.



*Acknowledgements.* This work has been supported by the Swiss National Science Foundation projects CHIMES (169676) and REUSE
(162668), by the European Union (H2020; ERC grant number 787574 PALAEO-RA), and by Euro-Climhist. We thank all the students who
helped in digitising the historical data.




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

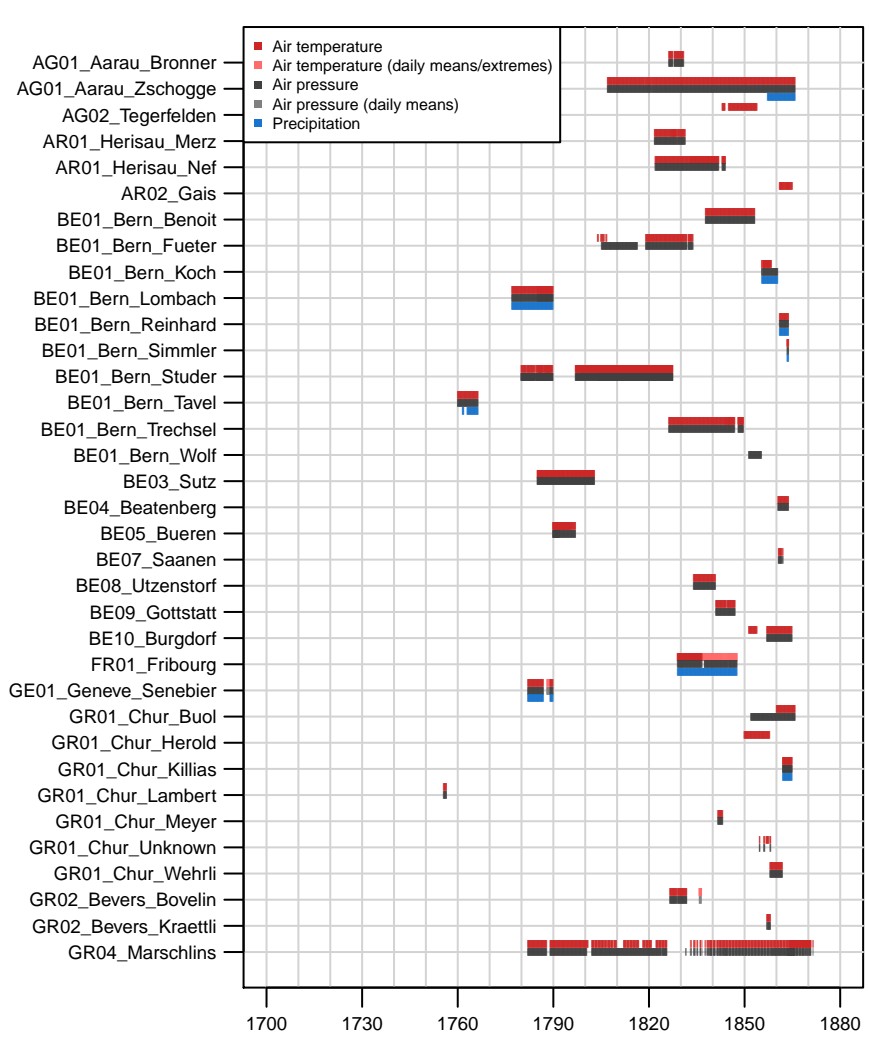

**Figure 1.** Availability of digitised data for the three core variables.

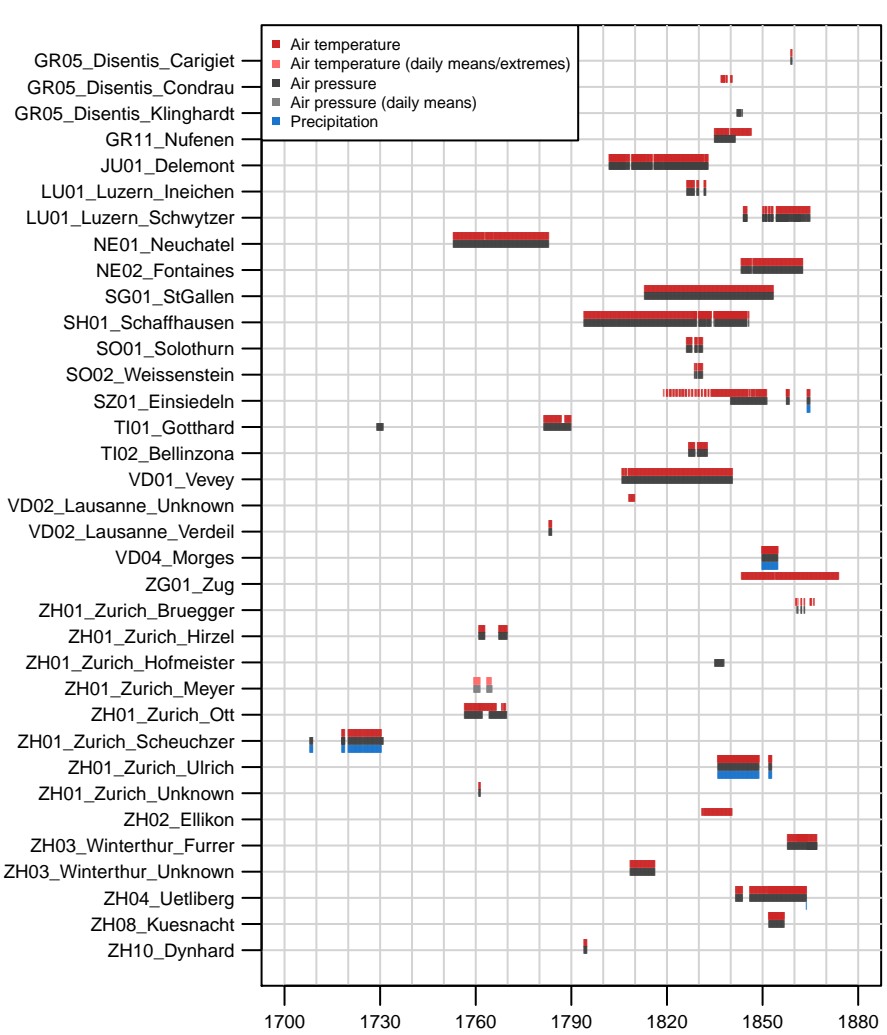

**Figure 1.** (continued)



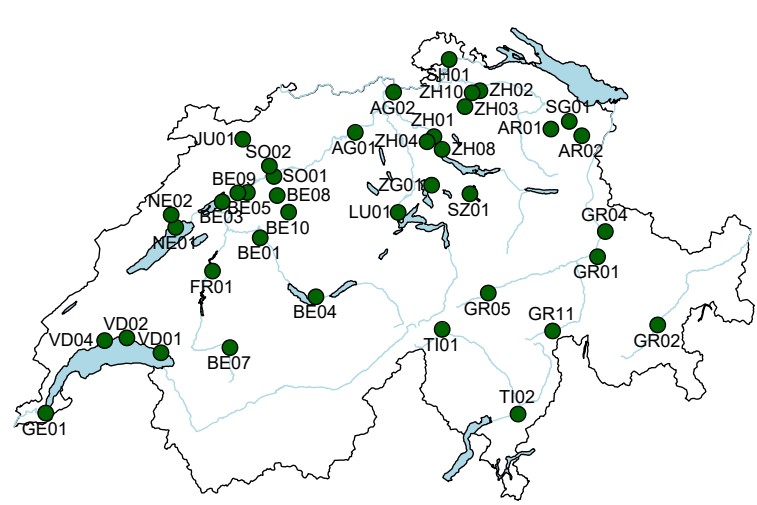

**Figure 2.** Map of the stations for which data have been digitised.

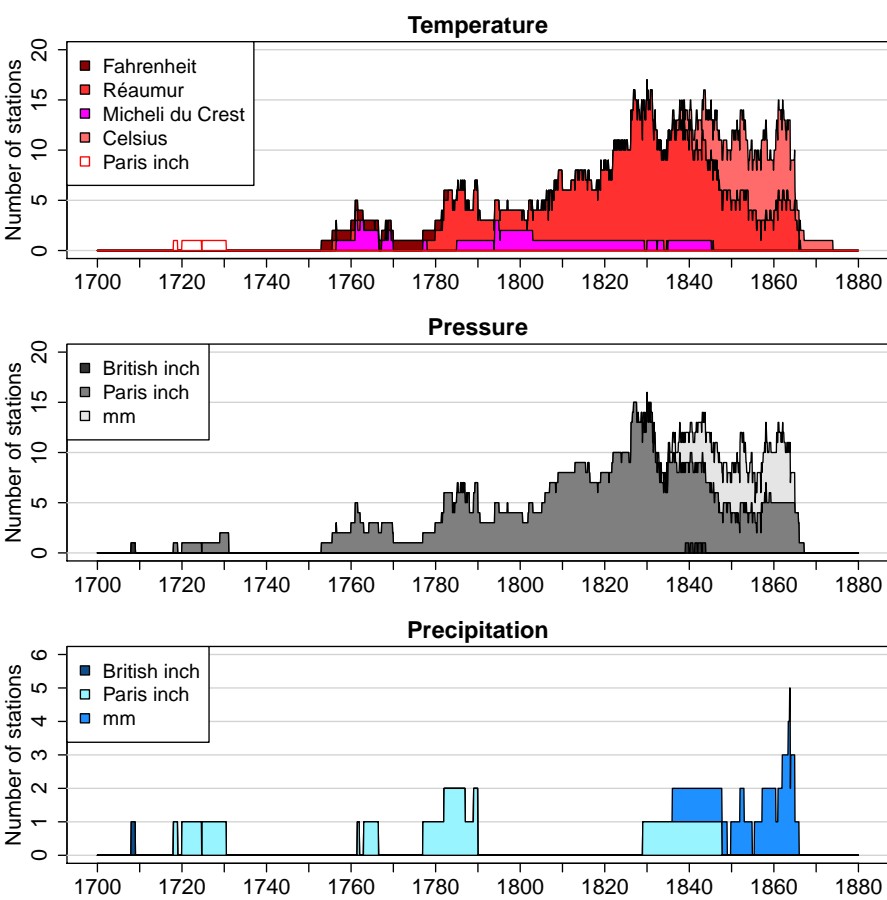

**Figure 3.** Temporal evolution of the number of stations for which data have been digitised. Colour shades indicate in which units the observations were reported.



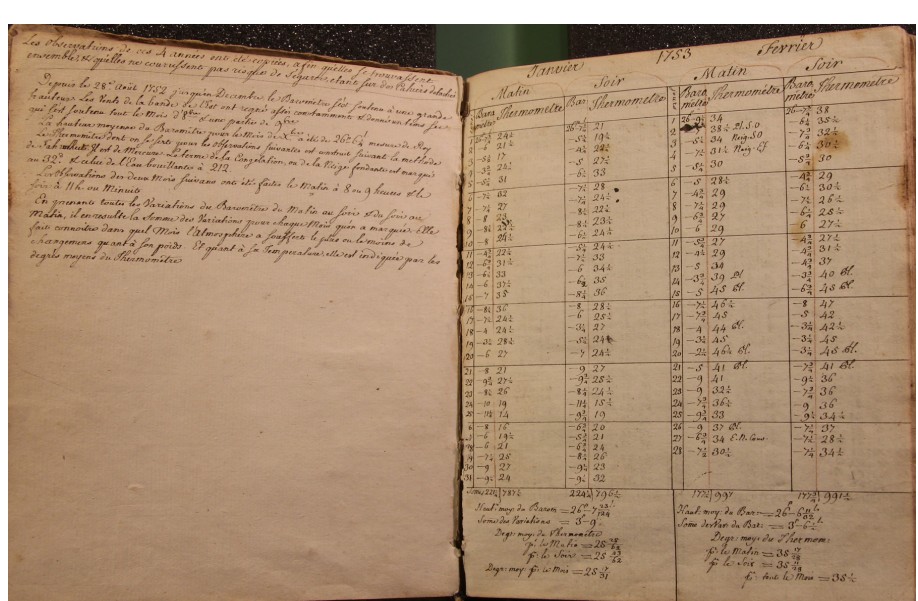

**Figure 4.** First page of Moula's meteorological register for the year 1753.



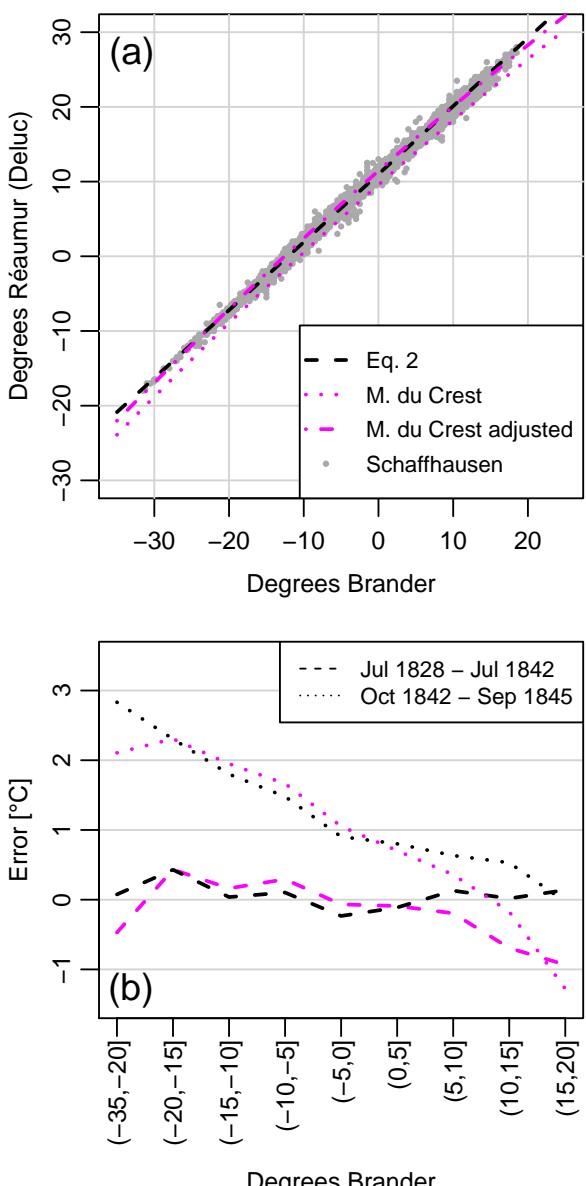

**Figure 5.** a) Scatter plot of parallel observations in Schaffhausen between a thermometer with Brander scale and one with Réaumur scale, made between 1828–1842; the lines represent different possible conversion functions discussed in the text. b) Median of differences between the parallel observations when using Eq. 2 (black lines) or a bias-adjusted Micheli du Crest scale (purple lines) for conversion, for different intervals of temperature.



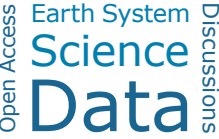

```
SEF      0.2.0
ID       BE01_Bern_Benoit
Name     Bern
Lat      46.94812
Lon      7.45196
Alt      534
Source   CHIMES
Link     https://doi.org/10.5194/cp-15-1345-2019
Vbl      p
Stat     point
Units    hPa
Meta     Observer=Daniel Gottlieb Benoît | PTC=N | PGC=Y
Year    Month   Day     Hour    Minute  Period  Value   Meta
1837    10      1       5       30      0       959.4   orig=26.7.0Pin | orig.time=6
1837    10      1       13      30      0       961.4   orig=26.7.8Pin | orig.time=14
1837    10      2       5       30      0       963.6   orig=26.8.5Pin | orig.time=6
1837    10      2       13      30      0       966.1   orig=26.9.3Pin | orig.time=14
1837    10      3       5       30      0       964.4   orig=26.8.8Pin | orig.time=6
1837    10      3       13      30      0       965.9   orig=26.9.2Pin | orig.time=14
1837    10      4       5       30      0       964.1   orig=26.8.7Pin | orig.time=6
1837    10      4       13      30      0       964.6   orig=26.8.9Pin | orig.time=14
1837    10      5       5       30      0       965.9   orig=26.9.2Pin | orig.time=6
1837    10      5       13      30      0       966.1   orig=26.9.3Pin | orig.time=14
1837    10      6       5       30      0       963.6   orig=26.8.5Pin | orig.time=6
1837    10      6       13      30      0       966.1   orig=26.9.3Pin | orig.time=14
1837    10      7       5       30      0       962.9   orig=26.8.2Pin | orig.time=6
1837    10      7       13      30      0       962.9   orig=26.8.2Pin | orig.time=14
```

**Figure 6.** Example of a SEF file for a pressure record.

**Table 1.** Number of digitised observations for each variable, with indication on whether they are provided as SEF files (observations in unknown units and redundant observations are not provided in SEF). Daily temperature observations include maximum, minimum, and mean temperature.

| Variable | Observations | SEF |
|---|---:|---|
| Temperature (sub-daily) | 877,048 | yes |
| Temperature (sub-daily) | 11,222 | no |
| Temperature (daily) | 45,516 | yes |
| Temperature (daily) | 31,436 | no |
| Pressure (sub-daily) | 744,291 | yes |
| Pressure (sub-daily) | 1,385 | no |
| Pressure (daily) | 31,785 | yes |
| Pressure (daily) | 1,949 | no |
| Precipitation | 62,598 | yes |
| Wind direction | 410,392 | no |
| Precipitation type/occurrence | 194,983 | no |
| Wind force | 83,561 | no |
| Humidity | 34,741 | no |
| Weather description | 22,161 | no |
| Fresh snow | 10,284 | no |
| Wet bulb temperature | 8,562 | no |
| Cloud cover | 3,497 | no |
| Water temperature | 904 | no |
| Soil temperature | 52 | no |
| **TOTAL** | **2,576,367** | |

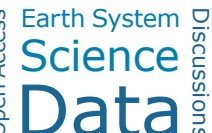

**Table 2.** Conversion table for the different temperature scales encountered in the dataset.

| °C  | Fahrenheit | Deluc (Réaumur) | Micheli du Crest | Brander |
|-----|------------|-----------------|------------------|---------|
| 35  | 95         | 28              | 22.0             | 18.7    |
| 30  | 86         | 24              | 17.0             | 14.3    |
| 25  | 77         | 20              | 12.2             | 9.9     |
| 20  | 68         | 16              | 7.4              | 5.5     |
| 15  | 59         | 12              | 2.8              | 1.1     |
| 10  | 50         | 8               | -1.8             | -3.3    |
| 5   | 41         | 4               | -6.2             | -7.7    |
| 0   | 32         | 0               | -10.6            | -12.1   |
| -5  | 23         | -4              | -14.8            | -16.5   |
| -10 | 14         | -8              | -19.0            | -20.9   |
| -15 | 5          | -12             | -23.2            | -25.3   |
| -20 | -4         | -16             | -27.2            | -29.7   |
| -25 | -13        | -20             | -31.2            | -34.1   |
| -30 | -22        | -24             | -35.1            | -38.5   |