# Peer review of "Early instrumental meteorological observations in Switzerland: 1708–1873"

_Earth System Science Data, 2019_

## Referee Comment (RC1) · Anonymous Referee #1 · 16 Jan 2020

General comments

The paper enlightens on a new historical database, unpublished so far, which partially covers a period of time prior to the institutionalization of meteorological observation in Switzerland, and of great climatological interest. This element is the main novelty, since the methodology applied in the quality control process has already been applied previously. Efforts made to document the various instruments and units of measurement used in Switzerland, the periods in which they were active, and the conversion to the modern system are also highly appreciated by this reviewer.

The paper becomes the natural continuity of the previous one by Pfister et al. (2019), which describes in detail the contents of the new documentary collection, and their potential. The present paper focuses on giving details on how typing was performed,

under what criteria and which quality control was applied. Detailed information is provided regarding the available metadata, and at all times the authors are transparent about the uncertainty of records (units of measurement, exposure), which later in a forthcoming paper (already announced), it will be relevant for homogeneity analysis.

Specific comments

- Line 50. It is mentioned that figure 2 refers to the "distribution of stations" when in fact the figure shows the cities where information is available. That is, for the same city there are up to 10 stations or series. The term "cities" or "locations" would be preferable.

- Line 222. The presence of "problematic packages" is mentioned but the reason why they are problematic is not commented on. It would be nice to state the reasons, if the same issues may be found in other later digitization initiatives.

- Line 220. On dataresqc it could be added that this is, at the moment, an absolute quality control, which works with the data in the series itself and does not contextualize the data with coexisting and close series.

- Line 230. Figures on the amount of erroneus data detected by datarescq is provided, but it is not discerned between errors from the original data or errors entered in the digitisation process. Is this information available? It would be good to know which one has a stronger weight. At the same time, it is not stated if during the typing process, any distinction is made between a record that is not available because it does not exist, or because it could not be read. Is this information detailed in the metadata file?

Technical corrections

- Line 36. Apparently, there is a grammatical error in the phrase "amount of records the we found". "The" should be a "that".

- Line 290 - References. The link to the DOI of the publication by Brugnara, does not work.

---

## Referee Comment (RC2) · Anonymous Referee #2 · 5 Feb 2020

This review is about the article "Early instrumental meteorological observations in Switzerland: 1708-1873" by Y. Brugnara et al. The authors digitized a large set of old hand-written meteorological observations from several obervation stations around Switzerland, and are describing their stragy and conversion problems along the way.

I consider this article very interesting and I also appreciate the work that was done by the authors and students to get this work done. The article is also well written and explained, so I have only a few comments and minor requests.

The first one is regarding Fig 1: I think you should reconsider your color choice here. It is hard to distinguish between the red (air temperature) and pink (air temperature (daily means)). Same for air pressure/air pressure (daily means). Especially, when you have short or broken intervals, like e.g. ZH01_Zuerich_Bruegger). I understand that

you want to keep the colors of similar variables close, but in this case it causes more confusion than insight. I would propose to use a wieder color scale here.

The same "color problem" applies to Figure 3: Without reading through the text it is hard for me to distingish between the different shades of red.

Page 4, line 104: I understand the plan to address the uncertainty in another paper. However, often the next paper takes a while to get published and the users of the data are left hanging with no uncertainty estimate. If you could give a benchmark or an estimate-range for this current dataset with respect to uncertainty, then it would help a lot. The user can take this number until you provide a better and more accurate estimate.

page 5, formula (1a/b) : How reliable do you consider these conversion formulas? Is there a reason why you chose second degree polynomial? It would help a lot, if you could provide an uncertainty estimate. Without graphical or tabular support, it is difficult to get a a feeling for this correction.

p 6. Formula (2)/Fig 5a) : I am not quite sure, if I see the advantage of Eq 2 to the adjusted M. du Crest. Do you have any mathematical support for Eq. 2? Like a lower mean deviation from the observation points? Did you make any statistical tests of your linear regression?

p 7, line 195 : You should mention here, that "corrected" pressure values are makred differentlty in the meta data. It gets only mentioned 2 pages later and its therefore easy to read over it.

4 Conclusions

This chapter is very short. I think, it could be expanded a little bit. it would be nice to have a short summary about possible sources of uncertainty. Users of the data will need some benchmark numbers, especially with respect to error estimates or quality assurances. Perhaps also some guidance, how to use the data, e.g. if it isib possible

to filter this data to get higher or lower accuracies.

My Regards.
* * *

---

## Author Response (AR1)

We thank the two anonymous referees for their time and their suggestions. Note that as a consequence of the minor changes in the methods, the addition of 3 years of data for Gotthard, and of additional quality control on the data, we uploaded a new version of the dataset (v1.1) under the same DOI.

Anonymous Referee #1

General comments

The paper enlightens on a new historical database, unpublished so far, which partially covers a period of time prior to the institutionalization of meteorological observation in Switzerland, and of great climatological interest. This element is the main novelty, since the methodology applied in the quality control process has already been applied previously. Efforts made to document the various instruments and units of measurement used in Switzerland, the periods in which they were active, and the conversion to the modern system are also highly appreciated by this reviewer.

The paper becomes the natural continuity of the previous one by Pfister et al. (2019),which describes in detail the contents of the new documentary collection, and their potential. The present paper focuses on giving details on how typing was performed, under what criteria and which quality control was applied. Detailed information is provided regarding the available metadata, and at all times the authors are transparent about the uncertainty of records (units of measurement, exposure), which later in a forthcoming paper (already announced), it will be relevant for homogeneity analysis.

Specific comments

- Line 50. It is mentioned that figure 2 refers to the "distribution of stations" when in fact the figure shows the cities where information is available. That is, for the same city there are up to 10 stations or series. The term "cities" or "locations" would be preferable.

We changed to "locations".

- Line 222. The presence of "problematic packages" is mentioned but the reason why they are problematic is not commented on. It would be nice to state the reasons, if the same issues may be found in other later digitization initiatives.

The main problems were related to digitization of negative temperature and of non-decimal pressure readings. Also systematic misinterpretation of the handwriting was common. We added a sentence.

- Line 220. On dataresqc it could be added that this is, at the moment, an absolute quality control, which works with the data in the series itself and does not contextualize the data with coexisting and close series.

We added a sentence.

- Line 230. Figures on the amount of erroneus data detected by datarescq is provided, but it is not discerned between errors from the original data or errors entered in the digitisation process. Is this information available? It would be good to know which one has a stronger weight. At the same time, it is not stated if during the typing process ,any distinction is made between a record that is not available because it does not exist, or because it could not be read. Is this information detailed in the metadata file?

The amount of (probable) errors in the original data that have been detected is given 5 lines below (3,990 or 0.2%). This is similar to the amount of digitization errors that we could detect and correct (about 4,000). These figures, however, consider only rather large errors that can be detected by automatic or semi-automatic qc tests.

There is not a specific metadata field that distinguishes between missing and unreadable. There is a field with annotations by the digitizer that usually contains this information in plain text (often in German though). Some students, however, preferred to use a color code in the Excel templates that was not read into a common format. Therefore, the information is currently not available for all records in the published dataset, although it was recorded in some form.

Technical corrections

- Line 36. Apparently, there is a grammatical error in the phrase "amount of records the we found". "The" should be a "that".

Thanks.

- Line 290 - References. The link to the DOI of the publication by Brugnara, does not work.

The DOI will be registered only after the paper is accepted. In fact we will release a version 1.1 of the dataset with enhanced quality control and amended conversions as described in the revised manuscript.

Anonymous Referee #2

This review is about the article "Early instrumental meteorological observations in Switzerland: 1708-1873" by Y. Brugnara et al. The authors digitized a large set of old hand-written meteorological observations from several obervation stations around Switzerland, and are describing their stragy and conversion problems along the way.

I consider this article very interesting and I also appreciate the work that was done by the authors and students to get this work done. The article is also well written and explained, so I have only a few comments and minor requests.

The first one is regarding Fig 1: I think you should reconsider your color choice here. It is hard to distinguish between the red (air temperature) and pink (air temperature(daily means)). Same for air pressure/air pressure (daily means). Especially, when you have short or broken intervals, like e.g. ZH01_Zuerich_Bruegger). I understand that you want to keep the colors of similar variables close, but in this case it causes more confusion than insight. I would propose to use a wieder color scale here.

The same "color problem" applies to Figure 3: Without reading through the text it is hard for me to distingish between the different shades of red.

We changed the colors.

Page 4, line 104: I understand the plan to address the uncertainty in another paper. However, often the next paper takes a while to get published and the users of the data are left hanging with no uncertainty estimate. If you could give a benchmark or an estimate-range for this current dataset with respect to uncertainty, then it would help a lot. The user can take this number until you provide a better and more accurate estimate.

Probably a quantitative estimation of the uncertainty is not really possible because of the general lack of metadata (in virtually all cases we do not know the exact specifics of the instruments nor their exposure), but we are publishing short articles about each record where we analyze in more detail (and in a standardized way) internal consistency and where we compare with nearby as well as modern records. From this articles users can get an idea of the uncertainty. Some of the articles have already been published (https://www.geography.unibe.ch/services/geographica_bernensia/online_publications/gb2020g96/index_eng.html) and we will mention them in the revised manuscript.
In addition, an important information on uncertainty that we provide is whether pressure is corrected for temperature or not.

page 5, formula (1a/b) : How reliable do you consider these conversion formulas? Is there a reason why you chose second degree polynomial? It would help a lot, if you could provide an uncertainty estimate. Without graphical or tabular support, it is difficult to get a a feeling for this correction.

We rewrote that section using slightly different formulas and adding more explanations.

p 6. Formula (2)/Fig 5a) : I am not quite sure, if I see the advantage of Eq 2 to the adjusted M. du Crest. Do you have any mathematical support for Eq. 2? Like a lower mean deviation from the observation points? Did you make any statistical tests of your linear regression?

The root-mean-square error is slightly better (difference of 0.02°C) for the linear formula, and more significantly the errors at the edges (low and high temperatures, where the second degree term makes the larger difference) are ~1°C lower on average (assuming that the parallel Réaumur thermometer is a reliable reference). Given that we do not know how the Brander thermometer was calibrated, it makes sense to take an empirical best fit.

p 7, line 195 : You should mention here, that "corrected" pressure values are makred differentlty in the meta data. It gets only mentioned 2 pages later and its therefore easy to read over it.

Done.

4 Conclusions

This chapter is very short. I think, it could be expanded a little bit. it would be nice to have a short summary about possible sources of uncertainty. Users of the data will need some benchmark numbers, especially with respect to error estimates or quality assurances. Perhaps also some guidance, how to use the data, e.g. if it isib possible to filter this data to get higher or lower accuracies.

We added a summary of possible sources of uncertainty with quantitative estimations from literature as well as some guidance.

[revised manuscript text omitted]